# Spectral Methods Meet EM: A Provably Optimal Algorithm for Crowdsourcing

**Yuchen Zhang**[†]     **Xi Chen**[♯]     **Dengyong Zhou**[∗]     **Michael I. Jordan**[†]

[†]University of California, Berkeley, Berkeley, CA 94720
{yuczhang,jordan}@berkeley.edu

[♯]New York University, New York, NY 10012
xichen@nyu.edu

[∗]Microsoft Research, 1 Microsoft Way, Redmond, WA 98052
dengyong.zhou@microsoft.com

## Abstract

The Dawid-Skene estimator has been widely used for inferring the true labels from the noisy labels provided by non-expert crowdsourcing workers. However, since the estimator maximizes a non-convex log-likelihood function, it is hard to theoretically justify its performance. In this paper, we propose a two-stage efficient algorithm for multi-class crowd labeling problems. The first stage uses the spectral method to obtain an initial estimate of parameters. Then the second stage refines the estimation by optimizing the objective function of the Dawid-Skene estimator via the EM algorithm. We show that our algorithm achieves the optimal convergence rate up to a logarithmic factor. We conduct extensive experiments on synthetic and real datasets. Experimental results demonstrate that the proposed algorithm is comparable to the most accurate empirical approach, while outperforming several other recently proposed methods.

## 1 Introduction

With the advent of online crowdsourcing services such as Amazon Mechanical Turk, crowdsourcing has become an appealing way to collect labels for large-scale data. Although this approach has virtues in terms of scalability and immediate availability, labels collected from the crowd can be of low quality since crowdsourcing workers are often non-experts and can be unreliable. As a remedy, most crowdsourcing services resort to labeling redundancy, collecting multiple labels from different workers for each item. Such a strategy raises a fundamental problem in crowdsourcing: how to infer true labels from noisy but redundant worker labels?

For labeling tasks with $k$ different categories, Dawid and Skene [8] propose a maximum likelihood approach based on the Expectation-Maximization (EM) algorithm. They assume that each worker is associated with a $k \times k$ confusion matrix, where the $(l, c)$-th entry represents the probability that a randomly chosen item in class $l$ is labeled as class $c$ by the worker. The true labels and worker confusion matrices are jointly estimated by maximizing the likelihood of the observed worker labels, where the unobserved true labels are treated as latent variables. Although this EM-based approach has had empirical success [21, 20, 19, 26, 6, 25], there is as yet no theoretical guarantee for its performance. A recent theoretical study [10] shows that the global optimal solutions of the Dawid-Skene estimator can achieve minimax rates of convergence in a simplified scenario, where the labeling task is binary and each worker has a single parameter to represent her labeling accuracy (referred to as a "one-coin model" in what follows). However, since the likelihood function is non-convex, this guarantee is not operational because the EM algorithm may get trapped in a local optimum. Several alternative approaches have been developed that aim to circumvent the theoretical deficiencies of the EM algorithm, still in the context of the one-coin model [14, 15, 11, 7]. Unfor-

tunately, they either fail to achieve the optimal rates or depend on restrictive assumptions which are hard to justify in practice.

We propose a computationally efficient and provably optimal algorithm to simultaneously estimate true labels and worker confusion matrices for multi-class labeling problems. Our approach is a two-stage procedure, in which we first compute an initial estimate of worker confusion matrices using the spectral method, and then in the second stage we turn to the EM algorithm. Under some mild conditions, we show that this two-stage procedure achieves minimax rates of convergence up to a logarithmic factor, even after only one iteration of EM. In particular, given any $\delta \in (0, 1)$, we provide the bounds on the number of workers and the number of items so that our method can correctly estimate labels for all items with probability at least $1 - \delta$. We also establish a lower bound to demonstrate the optimality of this approach. Further, we provide both upper and lower bounds for estimating the confusion matrix of each worker and show that our algorithm achieves the optimal accuracy.

This work not only provides an optimal algorithm for crowdsourcing but sheds light on understanding the general method of moments. Empirical studies show that when the spectral method is used as an initialization for the EM algorithm, it outperforms EM with random initialization [18, 5]. This work provides a concrete way to theoretically justify such observations. It is also known that starting from a root-$n$ consistent estimator obtained by the spectral method, one Newton-Raphson step leads to an asymptotically optimal estimator [17]. However, obtaining a root-$n$ consistent estimator and performing a Newton-Raphson step can be demanding computationally. In contrast, our initialization doesn't need to be root-$n$ consistent, thus a small portion of data suffices to initialize. Moreover, performing one iteration of EM is computationally more attractive and numerically more robust than a Newton-Raphson step especially for high-dimensional problems.

## 2   Related Work

Many methods have been proposed to address the problem of estimating true labels in crowdsourcing [23, 20, 22, 11, 19, 26, 7, 15, 14, 25]. The methods in [20, 11, 15, 19, 14, 7] are based on the generative model proposed by Dawid and Skene [8]. In particular, Ghosh et al. [11] propose a method based on Singular Value Decomposition (SVD) which addresses binary labeling problems under the one-coin model. The analysis in [11] assumes that the labeling matrix is full, that is, each worker labels all items. To relax this assumption, Dalvi et al. [7] propose another SVD-based algorithm which explicitly considers the sparsity of the labeling matrix in both algorithm design and theoretical analysis. Karger et al. propose an iterative algorithm for binary labeling problems under the one-coin model [15] and extend it to multi-class labeling tasks by converting a $k$-class problem into $k - 1$ binary problems [14]. This line of work assumes that tasks are assigned to workers according to a random regular graph, thus imposing specific constraints on the number of workers and the number of items. In Section 5, we compare our theoretical results with that of existing approaches [11, 7, 15, 14]. The methods in [20, 19, 6] incorporate Bayesian inference into the Dawid-Skene estimator by assuming a prior over confusion matrices. Zhou et al. [26, 25] propose a minimax entropy principle for crowdsourcing which leads to an exponential family model parameterized with worker ability and item difficulty. When all items have zero difficulty, the exponential family model reduces to the generative model suggested by Dawid and Skene [8].

Our method for initializing the EM algorithm in crowdsourcing is inspired by recent work using spectral methods to estimate latent variable models [3, 1, 4, 2, 5, 27, 12, 13]. The basic idea in this line of work is to compute third-order empirical moments from the data and then to estimate parameters by computing a certain orthogonal decomposition of a tensor derived from the moments. Given the special symmetric structure of the moments, the tensor factorization can be computed efficiently using the robust tensor power method [3]. A problem with this approach is that the estimation error can have a poor dependence on the condition number of the second-order moment matrix and thus empirically it sometimes performs worse than EM with multiple random initializations. Our method, by contrast, requires only a rough initialization from the moment of moments; we show that the estimation error does not depend on the condition number (see Theorem 2 (b)).

## 3   Problem Setup

Throughout this paper, $[a]$ denotes the integer set $\{1, 2, \ldots, a\}$ and $\sigma_b(A)$ denotes the $b$-th largest singular value of the matrix $A$. Suppose that there are $m$ workers, $n$ items and $k$ classes. The true

---

**Algorithm 1:** Estimating confusion matrices

---

**Input**: integer $k$, observed labels $z_{ij} \in \mathbb{R}^k$ for $i \in [m]$ and $j \in [n]$.

**Output**: confusion matrix estimates $\widehat{C}_i \in \mathbb{R}^{k \times k}$ for $i \in [m]$.

(1) Partition the workers into three disjoint and non-empty group $G_1$, $G_2$ and $G_3$. Compute the group aggregated labels $Z_{gj}$ by Eq. (1).

(2) For $(a, b, c) \in \{(2, 3, 1), (3, 1, 2), (1, 2, 3)\}$, compute the second and third order moments $\widehat{M}_2 \in \mathbb{R}^{k \times k}$, $\widehat{M}_3 \in \mathbb{R}^{k \times k \times k}$ by Eq. (2a)-(2d), then compute $\widehat{C}_c^\diamond \in \mathbb{R}^{k \times k}$ and $\widehat{W} \in \mathbb{R}^{k \times k}$ by tensor decomposition:

   (a) Compute whitening matrix $\widehat{Q} \in \mathbb{R}^{k \times k}$ (such that $\widehat{Q}^T \widehat{M}_2 \widehat{Q} = I$) using SVD.

   (b) Compute eigenvalue-eigenvector pairs $\{(\widehat{\alpha}_h, \widehat{v}_h)\}_{h=1}^k$ of the whitened tensor $\widehat{M}_3(\widehat{Q}, \widehat{Q}, \widehat{Q})$ by using the robust tensor power method [3]. Then compute $\widehat{w}_h = \widehat{\alpha}_h^{-2}$ and $\widehat{\mu}_h^\diamond = (\widehat{Q}^T)^{-1}(\widehat{\alpha}_h \widehat{v}_h)$.

   (c) For $l = 1, \ldots, k$, set the $l$-th column of $\widehat{C}_c^\diamond$ by some $\widehat{\mu}_h^\diamond$ whose $l$-th coordinate has the greatest component, then set the $l$-th diagonal entry of $\widehat{W}$ by $\widehat{w}_h$.

(3) Compute $\widehat{C}_i$ by Eq. (3).

---

label $y_j$ of item $j \in [n]$ is assumed to be sampled from a probability distribution $\mathbb{P}[y_j = l] = w_l$ where $\{w_l : l \in [k]\}$ are positive values satisfying $\sum_{l=1}^k w_l = 1$. Denote by a vector $z_{ij} \in \mathbb{R}^k$ the label that worker $i$ assigns to item $j$. When the assigned label is $c$, we write $z_{ij} = e_c$, where $e_c$ represents the $c$-th canonical basis vector in $\mathbb{R}^k$ in which the $c$-th entry is 1 and all other entries are 0. A worker may not label every item. Let $\pi_i$ indicate the probability that worker $i$ labels a randomly chosen item. If item $j$ is not labeled by worker $i$, we write $z_{ij} = 0$. Our goal is to estimate the true labels $\{y_j : j \in [n]\}$ from the observed labels $\{z_{ij} : i \in [m], j \in [n]\}$.

In order to obtain an estimator, we need to make assumptions on the process of generating observed labels. Following the work of Dawid and Skene [8], we assume that the probability that worker $i$ labels an item in class $l$ as class $c$ is independent of any particular chosen item, that is, it is a constant over $j \in [n]$. Let us denote the constant probability by $\mu_{ilc}$. Let $\mu_{il} = [\mu_{il1}\ \mu_{il2}\ \cdots\ \mu_{ilk}]^T$. The matrix $C_i = [\mu_{i1}\ \mu_{i2}\ \ldots\ \mu_{ik}] \in \mathbb{R}^{k \times k}$ is called the *confusion matrix* of worker $i$. Besides estimating the true labels, we also want to estimate the confusion matrix for each worker.

## 4   Our Algorithm

In this section, we present an algorithm to estimate confusion matrices and true labels. Our algorithm consists of two stages. In the first stage, we compute an initial estimate of confusion matrices via the method of moments. In the second stage, we perform the standard EM algorithm by taking the result of the Stage 1 as an initialization.

### 4.1   Stage 1: Estimating Confusion Matrices

Partitioning the workers into three disjoint and non-empty groups $G_1$, $G_2$ and $G_3$, the outline of this stage is the following: we use the spectral method to estimate the averaged confusion matrices for the three groups, then utilize this intermediate estimate to obtain the confusion matrix of each individual worker. In particular, for $g \in \{1, 2, 3\}$ and $j \in [n]$, we calculate the averaged labeling within each group by

$$Z_{gj} := \frac{1}{|G_g|} \sum_{i \in G_g} z_{ij}. \tag{1}$$

Denoting the aggregated confusion matrix columns by $\mu_{gl}^\diamond := \mathbb{E}(Z_{gj}|y_j = l) = \frac{1}{|G_g|} \sum_{i \in G_g} \pi_i \mu_{il}$, our first step is to estimate $C_g^\diamond := [\mu_{g1}^\diamond, \mu_{g2}^\diamond, \ldots, \mu_{gk}^\diamond]$ and to estimate the distribution of true labels

$W := \mathrm{diag}(w_1, w_2, \ldots, w_k)$. The following proposition shows that we can solve for $C_g^{\diamond}$ and $W$ from the moments of $\{Z_{gj}\}$.

**Proposition 1** (Anandkumar et al. [3]). *Assume that the vectors $\{\mu_{g1}^{\diamond}, \mu_{g2}^{\diamond}, \ldots, \mu_{gk}^{\diamond}\}$ are linearly independent for each $g \in \{1, 2, 3\}$. Let $(a, b, c)$ be a permutation of $\{1, 2, 3\}$. Define*

$$Z'_{aj} := \mathbb{E}[Z_{cj} \otimes Z_{bj}] \left(\mathbb{E}[Z_{aj} \otimes Z_{bj}]\right)^{-1} Z_{aj},$$

$$Z'_{bj} := \mathbb{E}[Z_{cj} \otimes Z_{aj}] \left(\mathbb{E}[Z_{bj} \otimes Z_{aj}]\right)^{-1} Z_{bj},$$

$$M_2 := \mathbb{E}[Z'_{aj} \otimes Z'_{bj}] \quad and \quad M_3 := \mathbb{E}[Z'_{aj} \otimes Z'_{bj} \otimes Z_{cj}];$$

*then we have $M_2 = \sum_{l=1}^{k} w_l \, \mu_{cl}^{\diamond} \otimes \mu_{cl}^{\diamond}$ and $M_3 = \sum_{l=1}^{k} w_l \, \mu_{cl}^{\diamond} \otimes \mu_{cl}^{\diamond} \otimes \mu_{cl}^{\diamond}$.*

Since we only have finite samples, the expectations in Proposition 1 have to be approximated by empirical moments. In particular, they are computed by averaging over indices $j = 1, 2, \ldots, n$. For each permutation $(a, b, c) \in \{(2, 3, 1), (3, 1, 2), (1, 2, 3)\}$, we compute

$$\widehat{Z}'_{aj} := \Big(\frac{1}{n}\sum_{j=1}^{n} Z_{cj} \otimes Z_{bj}\Big)\Big(\frac{1}{n}\sum_{j=1}^{n} Z_{aj} \otimes Z_{bj}\Big)^{-1} Z_{aj}, \tag{2a}$$

$$\widehat{Z}'_{bj} := \Big(\frac{1}{n}\sum_{j=1}^{n} Z_{cj} \otimes Z_{aj}\Big)\Big(\frac{1}{n}\sum_{j=1}^{n} Z_{bj} \otimes Z_{aj}\Big)^{-1} Z_{bj}, \tag{2b}$$

$$\widehat{M}_2 := \frac{1}{n}\sum_{j=1}^{n} \widehat{Z}'_{aj} \otimes \widehat{Z}'_{bj}, \tag{2c}$$

$$\widehat{M}_3 := \frac{1}{n}\sum_{j=1}^{n} \widehat{Z}'_{aj} \otimes \widehat{Z}'_{bj} \otimes Z_{cj}. \tag{2d}$$

The statement of Proposition 1 suggests that we can recover the columns of $C_c^{\diamond}$ and the diagonal entries of $W$ by operating on the moments $\widehat{M}_2$ and $\widehat{M}_3$. This is implemented by the tensor factorization method in Algorithm 1. In particular, the tensor factorization algorithm returns a set of vectors $\{(\widehat{\mu}_h^{\diamond}, \widehat{w}_h) : h = 1, \ldots, k\}$, where each $(\widehat{\mu}_h^{\diamond}, \widehat{w}_h)$ estimates a particular column of $C_c^{\diamond}$ (for some $\mu_{cl}^{\diamond}$) and a particular diagonal entry of $W$ (for some $w_l$). It is important to note that the tensor factorization algorithm doesn't provide a one-to-one correspondence between the recovered column and the true columns of $C_c^{\diamond}$. Thus, $\widehat{\mu}_1^{\diamond}, \ldots, \widehat{\mu}_k^{\diamond}$ represents an arbitrary permutation of the true columns.

To discover the index correspondence, we take each $\widehat{\mu}_h^{\diamond}$ and examine its greatest component. We assume that within each group, the probability of assigning a correct label is always greater than the probability of assigning any specific incorrect label. This assumption will be made precise in the next section. As a consequence, if $\widehat{\mu}_h^{\diamond}$ corresponds to the $l$-th column of $C_c^{\diamond}$, then its $l$-th coordinate is expected to be greater than other coordinates. Thus, we set the $l$-th column of $\widehat{C}_c^{\diamond}$ to some vector $\widehat{\mu}_h^{\diamond}$ whose $l$-th coordinate has the greatest component (if there are multiple such vectors, then randomly select one of them; if there is no such vector, then randomly select a $\widehat{\mu}_h^{\diamond}$). Then, we set the $l$-th diagonal entry of $\widehat{W}$ to the scalar $\widehat{w}_h$ associated with $\widehat{\mu}_h^{\diamond}$. Note that by iterating over $(a, b, c) \in \{(2, 3, 1), (3, 1, 2), (1, 2, 3)\}$, we obtain $\widehat{C}_c^{\diamond}$ for $c = 1, 2, 3$ respectively. There will be three copies of $\widehat{W}$ estimating the same matrix $W$—we average them for the best accuracy.

In the second step, we estimate each individual confusion matrix $C_i$. The following proposition shows that we can recover $C_i$ from the moments of $\{z_{ij}\}$. See [24] for the proof.

**Proposition 2.** *For any $g \in \{1, 2, 3\}$ and any $i \in G_g$, let $a \in \{1, 2, 3\}\backslash\{g\}$ be one of the remaining group index. Then*

$$\pi_i C_i W (C_a^{\diamond})^T = \mathbb{E}[z_{ij} Z_{aj}^T].$$

Proposition 2 suggests a plug-in estimator for $C_i$. We compute $\widehat{C}_i$ using the empirical approximation of $\mathbb{E}[z_{ij} Z_{aj}^T]$ and using the matrices $\widehat{C}_a^{\diamond}, \widehat{C}_b^{\diamond}, \widehat{W}$ obtained in the first step. Concretely, we calculate

$$\widehat{C}_i := \mathrm{normalize}\left\{\Big(\frac{1}{n}\sum_{j=1}^{n} z_{ij} Z_{aj}^T\Big)\Big(\widehat{W}(\widehat{C}_a^{\diamond})^T\Big)^{-1}\right\}, \tag{3}$$

where the normalization operator rescales the matrix columns, making sure that each column sums to one. The overall procedure for Stage 1 is summarized in Algorithm 1.

## 4.2 Stage 2: EM algorithm

The second stage is devoted to refining the initial estimate provided by Stage 1. The joint likelihood of true label $y_j$ and observed labels $z_{ij}$, as a function of confusion matrices $\mu_i$, can be written as

$$L(\mu; y, z) := \prod_{j=1}^{n} \prod_{i=1}^{m} \prod_{c=1}^{k} (\mu_{iy_jc})^{\mathbb{I}(z_{ij}=e_c)}.$$

By assuming a uniform prior over $y$, we maximize the marginal log-likelihood function $\ell(\mu) := \log(\sum_{y \in [k]^n} L(\mu; y, z))$. We refine the initial estimate of Stage 1 by maximizing the objective function, which is implemented by the Expectation Maximization (EM) algorithm. The EM algorithm takes the values $\{\widehat{\mu}_{ilc}\}$ provided as output by Stage 1 as initialization, then executes the following E-step and M-step *for at least one round*.

**E-step** Calculate the expected value of the log-likelihood function, with respect to the conditional distribution of $y$ given $z$ under the current estimate of $\mu$:

$$Q(\mu) := \mathbb{E}_{y|zf,\widehat{\mu}}\left[\log(L(\mu; y, z))\right] = \sum_{j=1}^{n}\left\{\sum_{l=1}^{k} \widehat{q}_{jl} \log\left(\prod_{i=1}^{m}\prod_{c=1}^{k}(\mu_{ilc})^{\mathbb{I}(z_{ij}=e_c)}\right)\right\},$$

$$\text{where} \quad \widehat{q}_{jl} \leftarrow \frac{\exp\left(\sum_{i=1}^{m}\sum_{c=1}^{k}\mathbb{I}(z_{ij}=e_c)\log(\widehat{\mu}_{ilc})\right)}{\sum_{l'=1}^{k}\exp\left(\sum_{i=1}^{m}\sum_{c=1}^{k}\mathbb{I}(z_{ij}=e_c)\log(\widehat{\mu}_{il'c})\right)} \quad \text{for } j \in [n], l \in [k].$$

$$(4)$$

**M-step** Find the estimate $\widehat{\mu}$ that maximizes the function $Q(\mu)$:

$$\widehat{\mu}_{ilc} \leftarrow \frac{\sum_{j=1}^{n}\widehat{q}_{jl}\mathbb{I}(z_{ij}=e_c)}{\sum_{c'=1}^{k}\sum_{j=1}^{n}\widehat{q}_{jl}\mathbb{I}(z_{ij}=e_{c'})} \quad \text{for } i \in [m], l \in [k], c \in [k]. \quad (5)$$

In practice, we alternatively execute the updates (4) and (5), for one iteration or until convergence. Each update increases the objective function $\ell(\mu)$. Since $\ell(\mu)$ is not concave, the EM update doesn't guarantee converging to the global maximum. It may converge to distinct local stationary points for different initializations. Nevertheless, as we prove in the next section, it is guaranteed that the EM algorithm will output statistically optimal estimates of true labels and worker confusion matrices if it is initialized by Algorithm 1.

## 5 Convergence Analysis

To state our main theoretical results, we first need to introduce some notation and assumptions. Let

$$w_{\min} := \min\{w_l\}_{l=1}^{k} \quad \text{and} \quad \pi_{\min} := \min\{\pi_i\}_{i=1}^{m}$$

be the smallest portion of true labels and the most extreme sparsity level of workers. Our first assumption assumes that both $w_{\min}$ and $\pi_{\min}$ are strictly positive, that is, every class and every worker contributes to the dataset.

Our second assumption assumes that the confusion matrices for each of the three groups, namely $C_1^\diamond$, $C_2^\diamond$ and $C_3^\diamond$, are nonsingular. As a consequence, if we define matrices $S_{ab}$ and tensors $T_{abc}$ for any $a, b, c \in \{1, 2, 3\}$ as

$$S_{ab} := \sum_{l=1}^{k} w_l\, \mu_{al}^\diamond \otimes \mu_{bl}^\diamond = C_a^\diamond W(C_b^\diamond)^T \quad \text{and} \quad T_{abc} := \sum_{l=1}^{k} w_l\, \mu_{al}^\diamond \otimes \mu_{bl}^\diamond \otimes \mu_{cl}^\diamond,$$

then there will be a positive scalar $\sigma_L$ such that $\sigma_k(S_{ab}) \geq \sigma_L > 0$.

Our third assumption assumes that within each group, the average probability of assigning a correct label is always higher than the average probability of assigning any incorrect label. To make this

statement rigorous, we define a quantity

$$\kappa := \min_{g\in\{1,2,3\}} \min_{l\in[k]} \min_{c\in[k]\setminus\{l\}} \{\mu_{gll}^{\diamond} - \mu_{glc}^{\diamond}\}$$

indicating the smallest gap between diagonal entries and non-diagonal entries in the same confusion matrix column. The assumption requires $\kappa$ being strictly positive. Note that this assumption is group-based, thus does not assume the accuracy of any individual worker.

Finally, we introduce a quantity that measures the average ability of workers in identifying distinct labels. For two discrete distributions $P$ and $Q$, let $\mathbb{D}_{\mathrm{KL}}(P,Q) := \sum_i P(i)\log(P(i)/Q(i))$ represent the KL-divergence between $P$ and $Q$. Since each column of the confusion matrix represents a discrete distribution, we can define the following quantity:

$$\overline{D} = \min_{l\neq l'} \frac{1}{m}\sum_{i=1}^{m} \pi_i \mathbb{D}_{\mathrm{KL}}(\mu_{il}, \mu_{il'}). \tag{6}$$

The quantity $\overline{D}$ lower bounds the averaged KL-divergence between two columns. If $\overline{D}$ is strictly positive, it means that every pair of labels can be distinguished by at least one subset of workers. As the last assumption, we assume that $\overline{D}$ is strictly positive.

The following two theorems characterize the performance of our algorithm. We split the convergence analysis into two parts. Theorem 1 characterizes the performance of Algorithm 1, providing sufficient conditions for achieving an arbitrarily accurate initialization. We provide the proof of Theorem 1 in the long version of this paper [24].

**Theorem 1.** *For any scalar $\delta > 0$ and any scalar $\epsilon$ satisfying $\epsilon \leq \min\left\{\frac{36\kappa k}{\pi_{\min}w_{\min}\sigma_L}, 2\right\}$, if the number of items $n$ satisfies*

$$n = \Omega\left(\frac{k^5\log((k+m)/\delta)}{\epsilon^2\pi_{\min}^2 w_{\min}^2 \sigma_L^{13}}\right),$$

*then the confusion matrices returned by Algorithm 1 are bounded as*

$$\|\widehat{C}_i - C_i\|_{\infty} \leq \epsilon \qquad \text{for all } i \in [m],$$

*with probability at least $1-\delta$. Here, $\|\cdot\|_{\infty}$ denotes the element-wise $\ell_{\infty}$-norm of a matrix.*

Theorem 2 characterizes the error rate in Stage 2. It states that when a sufficiently accurate initialization is taken, the updates (4) and (5) refine the estimates $\widehat{\mu}$ and $\widehat{y}$ to the optimal accuracy. See the long version of this paper [24] for the proof.

**Theorem 2.** *Assume that there is a positive scalar $\rho$ such that $\mu_{ilc} \geq \rho$ for all $(i,l,c) \in [m]\times[k]^2$. For any scalar $\delta > 0$, if confusion matrices $\widehat{C}_i$ are initialized in a manner such that*

$$\|\widehat{C}_i - C_i\|_{\infty} \leq \alpha := \min\left\{\frac{\rho}{2}, \frac{\rho\overline{D}}{16}\right\} \qquad \text{for all } i \in [m], \tag{7}$$

*and the number of workers $m$ and the number of items $n$ satisfy*

$$m = \Omega\left(\frac{\log(1/\rho)\log(kn/\delta) + \log(mn)}{\overline{D}}\right) \quad \text{and} \quad n = \Omega\left(\frac{\log(mk/\delta)}{\pi_{\min}w_{\min}\alpha^2}\right),$$

*then, for $\widehat{\mu}$ and $\widehat{q}$ obtained by iterating (4) and (5) (for at least one round), with probability at least $1-\delta$,*

*(a) Letting $\widehat{y}_j = \arg\max_{l\in[k]}\widehat{q}_{jl}$, we have that $\widehat{y}_j = y_j$ holds for all $j \in [n]$.*

*(b) $\|\widehat{\mu}_{il} - \mu_{il}\|_2^2 \leq \frac{48\log(2mk/\delta)}{\pi_i w_l n}$ holds for all $(i,l) \in [m]\times[k]$.*

In Theorem 2, the assumption that all confusion matrix entries are lower bounded by $\rho > 0$ is somewhat restrictive. For datasets violating this assumption, we enforce positive confusion matrix entries by adding random noise: Given any observed label $z_{ij}$, we replace it by a random label in $\{1,...,k\}$ with probability $k\rho$. In this modified model, every entry of the confusion matrix is lower bounded by $\rho$, so that Theorem 2 holds. The random noise makes the constant $\overline{D}$ smaller than its original value, but the change is minor for small $\rho$.

| Dataset name | # classes | # items | # workers | # worker labels |
|:---:|:---:|:---:|:---:|:---:|
| Bird | 2 | 108 | 39 | 4,212 |
| RTE | 2 | 800 | 164 | 8,000 |
| TREC | 2 | 19,033 | 762 | 88,385 |
| Dog | 4 | 807 | 52 | 7,354 |
| Web | 5 | 2,665 | 177 | 15,567 |

Table 1: Summary of datasets used in the real data experiment.

To see the consequence of the convergence analysis, we take error rate $\epsilon$ in Theorem 1 equal to the constant $\alpha$ defined in Theorem 2. Then we combine the statements of the two theorems. This shows that if we choose the number of workers $m$ and the number of items $n$ such that

$$m = \widetilde{\Omega}\left(\frac{1}{\overline{D}}\right) \quad \text{and} \quad n = \widetilde{\Omega}\left(\frac{k^5}{\pi_{\min}^2 w_{\min}^2 \sigma_L^{13} \min\{\rho^2, (\rho\overline{D})^2\}}\right); \quad (8)$$

that is, if both $m$ and $n$ are lower bounded by a problem-specific constant and logarithmic terms, then with high probability, the predictor $\widehat{y}$ will be perfectly accurate, and the estimator $\widehat{\mu}$ will be bounded as $\|\widehat{\mu}_{il} - \mu_{il}\|_2^2 \leq \widetilde{\mathcal{O}}(1/(\pi_i w_l n))$. To show the optimality of this convergence rate, we present the following minimax lower bounds. Again, see [24] for the proof.

**Theorem 3.** *There are universal constants $c_1 > 0$ and $c_2 > 0$ such that:*

*(a) For any $\{\mu_{ilc}\}$, $\{\pi_i\}$ and any number of items $n$, if the number of workers $m \leq 1/(4\overline{D})$, then*

$$\inf_{\widehat{y}} \sup_{v \in [k]^n} \mathbb{E}\left[\sum_{j=1}^{n} \mathbb{I}(\widehat{y}_j \neq y_j)\Big|\{\mu_{ilc}\}, \{\pi_i\}, y = v\right] \geq c_1 n.$$

*(b) For any $\{w_l\}$, $\{\pi_i\}$, any worker-item pair $(m, n)$ and any pair of indices $(i, l) \in [m] \times [k]$, we have*

$$\inf_{\widehat{\mu}} \sup_{\mu \in \mathbb{R}^{m \times k \times k}} \mathbb{E}\left[\|\widehat{\mu}_{il} - \mu_{il}\|_2^2\Big|\{w_l\}, \{\pi_i\}\right] \geq c_2 \min\left\{1, \frac{1}{\pi_i w_l n}\right\}.$$

In part (a) of Theorem 3, we see that the number of workers should be at least $1/(4\overline{D})$, otherwise any predictor will make many mistakes. This lower bound matches our sufficient condition on the number of workers $m$ (see Eq. (8)). In part (b), we see that the best possible estimate for $\mu_{il}$ has $\Omega(1/(\pi_i w_l n))$ mean-squared error. It verifies the optimality of our estimator $\widehat{\mu}_{il}$. It is worth noting that the constraint on the number of items $n$ (see Eq. (8)) might be improvable. In real datasets we usually have $n \gg m$ so that the optimality for $m$ is more important than for $n$.

It is worth contrasting our convergence rate with existing algorithms. Ghosh et al. [11] and Dalvi et al. [7] proposed consistent estimators for the binary one-coin model. To attain an error rate $\delta$, their algorithms require $m$ and $n$ scaling with $1/\delta^2$, while our algorithm only requires $m$ and $n$ scaling with $\log(1/\delta)$. Karger et al. [15, 14] proposed algorithms for both binary and multi-class problems. Their algorithm assumes that workers are assigned by a random regular graph. Moreover, their analysis assumes that the limit of number of items goes to infinity, or that the number of workers is many times the number of items. Our algorithm no longer requires these assumptions.

We also compare our algorithm with the majority voting estimator, where the true label is simply estimated by a majority vote among workers. Gao and Zhou [10] showed that if there are many spammers and few experts, the majority voting estimator gives almost a random guess. In contrast, our algorithm only requires $m\overline{D} = \widetilde{\Omega}(1)$ to guarantee good performance. Since $m\overline{D}$ is the aggregated KL-divergence, a small number of experts are sufficient to ensure it is large enough.

## 6 Experiments

In this section, we report the results of empirical studies comparing the algorithm we propose in Section 4 (referred to as Opt-D&S) with a variety of existing methods which are also based on the generative model of Dawid and Skene. Specifically, we compare to the Dawid & Skene estimator

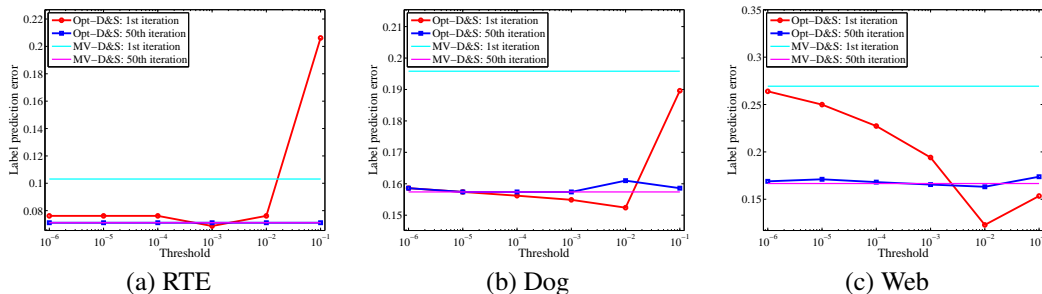

|       | (a) RTE | (b) Dog | (c) Web |
|-------|---------|---------|---------|

Figure 1: Comparing MV-D&S and Opt-D&S with different thresholding parameter $\Delta$. The label prediction error is plotted after the 1st EM update and after convergence.

|      | Opt-D&S | MV-D&S | Majority Voting | KOS | Ghosh-SVD | EigenRatio |
|------|---------|--------|-----------------|------|-----------|------------|
| Bird | **10.09** | 11.11 | 24.07 | 11.11 | 27.78 | 27.78 |
| RTE  | **7.12** | **7.12** | 10.31 | 39.75 | 49.13 | 9.00 |
| TREC | **29.80** | 30.02 | 34.86 | 51.96 | 42.99 | 43.96 |
| Dog  | 16.89 | **16.66** | 19.58 | 31.72 | – | – |
| Web  | 15.86 | **15.74** | 26.93 | 42.93 | – | – |

Table 2: Error rate (%) in predicting true labels on real data.

initialized by majority voting (referred to as MV-D&S), the pure majority voting estimator, the multi-class labeling algorithm proposed by Karger et al. [14] (referred to as KOS), the SVD-based algorithm proposed by Ghosh et al. [11] (referred to as Ghost-SVD) and the "Eigenvalues of Ratio" algorithm proposed by Dalvi et al. [7] (referred to as EigenRatio). The evaluation is made on five real datasets.

We compare the crowdsourcing algorithms on three binary tasks and two multi-class tasks. Binary tasks include labeling bird species [22] (Bird dataset), recognizing textual entailment [21] (RTE dataset) and assessing the quality of documents in the TREC 2011 crowdsourcing track [16] (TREC dataset). Multi-class tasks include labeling the breed of dogs from ImageNet [9] (Dog dataset) and judging the relevance of web search results [26] (Web dataset). The statistics for the five datasets are summarized in Table 1. Since the Ghost-SVD algorithm and the EigenRatio algorithm work on binary tasks, they are evaluated only on the Bird, RTE and TREC datasets. For the MV-D&S and the Opt-D&S methods, we iterate their EM steps until convergence.

Since entries of the confusion matrix are positive, we find it helpful to incorporate this prior knowledge into the initialization stage of the Opt-D&S algorithm. In particular, when estimating the confusion matrix entries by Eq. (3), we add an extra checking step before the normalization, examining if the matrix components are greater than or equal to a small threshold $\Delta$. For components that are smaller than $\Delta$, they are reset to $\Delta$. The default choice of the thresholding parameter is $\Delta = 10^{-6}$. Later, we will compare the Opt-D&S algorithm with respect to different choices of $\Delta$. It is important to note that this modification doesn't change our theoretical result, since the thresholding is not needed in case that the initialization error is bounded by Theorem 1.

Table 2 summarizes the performance of each method. The MV-D&S and the Opt-D&S algorithms consistently outperform the other methods in predicting the true label of items. The KOS algorithm, the Ghost-SVD algorithm and the EigenRatio algorithm yield poorer performance, presumably due to the fact that they rely on idealized assumptions that are not met by the real data. In Figure 1, we compare the Opt-D&S algorithm with respect to different thresholding parameters $\Delta \in \{10^{-i}\}_{i=1}^{6}$. We plot results for three datasets (RET, Dog, Web), where the performance of MV-D&S is equal to or slightly better than that of Opt-D&S. The plot shows that the performance of the Opt-D&S algorithm is stable after convergence. But at the first EM iterate, the error rates are more sensitive to the choice of $\Delta$. A proper choice of $\Delta$ makes Opt-D&S outperform MV-D&S. The result suggests that a proper initialization combined with one EM iterate is good enough for the purposes of prediction. In practice, the best choice of $\Delta$ can be obtained by cross validation.

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
