[Reviews · NeurIPS 2014]

Submitted by Assigned_Reviewer_15

This paper provides an application of the tensor decomposition method on a crowdsourcing problem. The setup is the following: there are m workers labeling n items for a set of k labels. We observe the labels z_{i, j} in {1...k} for i=1...m and j=1...n. We want to infer the true labels y_{j} for j=1...n, where the true labels are generated by a set of multi-view models corresponding to the workers.

The proposed recovery of the underlying model parameters is a straightforward application of the tensor decomposition method for multi-view models (Anandkumar et al., 2012). The authors also perform a local search via EM following the spectral method: initializing EM with spectral estimates have been shown to be effective (Chaganty and Liang, 2013), as acknowledged by the authors.

Notably, however, the authors analyze the maximum-likelihood estimates of EM in this setting and show that they recover the true labels whp. They also show the optimality of the convergence rate with a minimax argument.

To my knowledge, an attempt to analyze the behavior of EM initialized with spectral estimates is novel, and I think it's a fruitful and worthwhile direction.

Specific comments:

- It's probably helpful to more explicitly account for why you need the EM step for the results in Theorem 2. Wouldn't it be possible to argue similar results directly on the spectral estimates? If not, why?

- On a related note, the experiments are missing an obvious baseline: using the output of the spectral stage. It's probably much worse than following up with EM, but it's an important experiment.

- Change "(l, c)-th" to "(c, l)-th" in Introduction.

- As for the theshold delta, smoothing might be a cleaner solution and may give similar effects.

- A conclusion is needed.

Post-feedback comments:

- I understand the spectral method depends on various problem-specific constants, but it's still not clear to me why "its convergence rate is sub-optimal". Please clarify.

- I was suggesting an experiment of performing only the E-step on the spectral estimates. I'm assuming that Opt-D&S: 1st iteration in Figure 1 still performs an entire iteration of EM followed by another E-Step?
Summary: This paper provides an application of the tensor decomposition method and EM to a crowdsourcing problem. It also presents an analysis of estimation consistency and convergence optimality.

Submitted by Assigned_Reviewer_17

Thank you to the authors for their thoughtful feedback. I agree that there are pros and cons to both method-of-moments and optimization methods, and a detailed discussion of them is not needed in this paper (which, after all, aims to unite them!). But since the goal is to analyze an algorithm for a specific, practical problem, I think it is important to at least make sure the assumptions -- the main one being that the Dawid-Skene model is the process actually generating the data -- are clear so that a reader can decide whether they are plausible for a given application.

Finally, though the dependence on m may sometimes be more important than the dependence on n, I think it is important to explain this limitation when discussing optimality in the abstract and introduction, where the claims currently seem overstated. Even weakened in this way, they will still be interesting claims backed up by strong results.

--previous review below--

In this work the authors propose a two-stage method for extracting
consensus labels from the noisy ones assigned by possibly unreliable
annotators. In the first stage, the method applies spectral
techniques under the Dawid-Skene model of label noise to obtain
estimates of each annotator's label confusion matrix. It then uses
these estimates to initialize an EM algorithm, which alternately
updates the label estimates and the confusion matrices to improve
likelihood. The idea to initialize EM with spectral methods is not
new, but in this paper the authors claim to show that the result of
the two-stage process is near-optimal in an asymptotic sense.
Finally, experiments on several real-world datasets show that the
proposed method performs comparably to the best existing method.

As noted in the introduction, EM algorithms (like the one by Dawid and
Skene) generally set out to optimize non-convex likelihood functions,
which, being hard, makes it difficult to obtain any kind of
performance guarantees. The move to spectral methods allows for
results like Theorem 1, which shows that, given enough data, parameter
estimates will converge to the true parameters. However, there is a
tradeoff here, which is not mentioned: the guarantee only applies if
the model is correct. That is, if annotators have different confusion
tendencies on different items, or if their performance drifts over
time, or if they in any other way deviate from the Dawid-Skene model,
then Theorem 1 will likely not hold (nor will Theorem 2, in fact).
This problem is much less severe for optimization algorithms, like EM,
that have the ability to try and fit an imperfect model to real data;
they may not have global optimality guarantees, but their objectives
continue to guide them in settings where spectral theory simply fails.

The absence of this consideration is at the core of my biggest concern
about the paper. In attempting to prove the qualifications of a
method for handling the unreliability and unpredictability of third
party annotators, the paper assumes a high degree of regularity in
their behavior. To paraphrase a saying, some models are useful, but
all of them are wrong; I have mixed feelings about a theory that
assumes otherwise. At the very least, a thoughtful discussion of this
issue would be a valuable addition to the paper.

Having said all of that, I am excited about the direction the authors
of this paper have pursued. In practice, the spectral+EM strategy has
become popular; presumably the idea is to get a "best of both worlds"
kind of result, where the spectral method hopefully gives a good
initialization for EM, saving time and improving the final result, but
in any case is probably no worse than random initialization. Because
of this I find the idea of theory for the combined algorithm highly
interesting. I'm not sure, however, whether Theorem 2 really fulfills
this promise. In particular, the assumption that the input C matrices
are within alpha of the truth (for alpha that is small compared to the
truth) seems to require that the first stage has done basically all of
the work. In fact, the E-step used to obtain result (a) of Theorem 2
is just choosing the maximum-likelihood y using the given C. Arguably
EM is not really needed at all for the theory.

On the other hand, the experiments almost make the case for the
opposite possibility: that EM does most of the work. Whether
initialized by the spectral method or simple majority voting, the two
EM methods perform about the same. In some cases the spectral
initialization leads to immediate convergence, suggesting it has found
a good initialization, but even when the initialization is bad, 50
iterations of EM seem to bring it in line. Perhaps, then, the most
practical use of this method is as a time-saving technique; it would
be interesting to see the runtimes in that case.

I found Theorem 3 a bit hard to parse; presumably the \hat y and \hat
mu in the infima are supposed to be mappings from the observed data to
the quantity in question? At first glance they appear to just be
concrete values, which makes the claims seem a little strange. In any
case, I found the theorem itself interesting, but I'm not sure I agree
that it shows that the convergence rate is "near optimal" since it
doesn't address n, which is usually the default convergence clock and
arguably has the more dubious scaling in Equation 9, with large powers
of k and a hard-to-interpret singular value. In my opinion this makes
the optimality claims in the abstract and intro somewhat overstated.
Summary: Overall, I have mixed feelings about this paper. On the one hand, the
work tackles an interesting (and difficult) problem, managing to
successfully bring together several different types of theory as well
as fairly extensive empirical analysis. On the other hand, I'm not
convinced that the theory is as useful as it is made out to be, both
because the assumptions are strong, and because the lower bounds do
not consider n. The experiments suggest that the biggest benefit
might be in runtime, not error rate.

Submitted by Assigned_Reviewer_37

The authors propose a two-stage algorithm for estimating the parameters of a Dawid-Skene estimator (used for inferring true labels from noisy crowdsourced data). The algorithm relies on initializing EM using the method-of-moments (MoM), where the MoM portion of the estimation is performed via spectral (i.e., tensor) decomposition. To my knowledge, this is the first work to provide (relatively) strong optimality guarantees on such a combination. Work by Chagnaty et al in ICML 2014, for example, provides weaker guarantees on combining *composite* likelihood EM with spectral learning (though that work considers a more general latent-variable context). The authors show not only that the MoM-initialized EM algorithm converges to a globally optimal solution, they also show that the convergence rate is optimal in a minimax sense. These results do rely on a rather restrictive assumption (see comment 1 below), and the "problem specific constants" in the bound (i.e., those that are ignored in the comparison to the optimal minimax rates) are quite large (see comment 2 below). Nevertheless, the theoretical analysis is quite thorough, potentially providing insights for how MoM-EM hybrids can be analyzed in more general contexts, and the empirical results demonstrate that the algorithm is competitive with other state-of-the-art techniques, outperforming the alternatives in a significant proportion of cases.

Overall, this is a well-written paper and is a significant contribution to both the crowdsourcing and spectral learning communities. At a high level, I would like to see more discussion of what aspects of this particular problem made the elucidated bounds obtainable (i.e., what specific properties of this problem were crucial in proving the optimality of the MoM-EM combination), as this combination is not provably optimal in general latent-variable settings, but I understand that space does not necessarily permit such a discussion.

Comments:

1. The theoretical results assume that all entries of the worker confusion matrices are greater than some positive constant, $\rho$. This seems to a be *very* restrictive assumption in that it assumes that the workers always have some positive probability of confusing any two labels and thus seems to imply that the elucidated convergence rates will only hold in relatively "noisy" labeling problems (i.e., where all pairs of labels may be "mixed up" by workers). I think that the results are still compelling, despite this strong assumption; however, I think that the implications of the assumption should made more explicit via a brief comment in the primary text. For example, what proportion of crowdsourcing applications can be expected to satisfy this assumption? Do the authors think that the presented real-data experiments satisfy this assumption? (If not then this is strong evidence that the assumption is not strictly necessary for the empirical performance of the algorithm).

2. In the bound on $n$ in equation (9), the sample-complexity increases rapidly as the $\sigma_L$ decreases, and the constant $\rho$ also appears in the denominator. I think that a brief mention of the impact of these terms in the primary text would be useful. Essentially, it seems that the bounds require that $S_{a,b}$ is quite well-conditioned (otherwise $\sigma^13_L$ could be extremely small) and that workers are likely to mix up all pairwise label combinations, e.g. that there are not two labels that are almost never mixed up (otherwise $\rho$ could be extremely small). A brief comment on these terms could simultaneously address the question posed in Comment 1 as well. Even a simple intuitive explanation of what sorts of crowdsourcing applications can be expected to be "well-behaved" with respect to these constants would be informative.

Minor Comments:

- Line 260: Missing "$\min$" in definition of $\pi_{\min}$.

- In Table 2, for the the RTE data, the entry for OPT-D&S is bolded but the value is the same as for MV-D&S. Both or neither should be bolded.

Post-Feedback Comments:

W.r.t comment 1 above, authors pointed out that this problem can be alleviated by adding a small amount of noise to worker's labelings. This is certainly true, though it leads to a somewhat odd situation where adding noise is necessary to make the theory work by ensuring a positive $\rho$ but also degrades performance by decreasing $\bar{D}$. Essentially there is a tradeoff between $\rho$ and $\bar{D}$ (as larger $\rho$'s improve the bound for $n$) and it would be interesting if the author's could comment on a heuristic choice for the amount of noise to add, given this tradeoff. Also, is this perturbation actually necessary in practice (e.g., did the authors perform such a perturbation in the empirical analysis)?

W.r.t comment 2, the authors have agreed to address this issue via some more comments in the final draft.

As a final note, an important point that came up during discussion of the paper was that the claims of optimality, though well-founded, somewhat obscure an important issue. Specifically, the optimality of convergence is w.r.t. $m$ and not $n$, whereas it is perhaps more standard for $n$ to be used. The authors gave reasonable justification for this in their response, but the paper would be strengthened by the addition of such an explicit, brief justification.
Summary: This well-written paper presents a compelling application of the spectral method of moments with thorough empirical and theoretical analyses. It should be of interest to researchers in both the fields of crowdsourcing and spectral learning.
Author Feedback
Author rebuttal: We thank all reviewers for their careful reading and constructive comments.

Assigned_Reviewer_15

The EM step is essential here to achieve the optimal rate. As shown in Theorem 1, the spectral method provides a consistent estimator, but its sample complexity depends on a large problem-specific constant such that its convergence rate is sub-optimal. In our algorithm, we first use the spectral method to obtain an $\alpha$-accurate initialization (Eq. (8)), and then employ EM to boost it to optimal (Theorem 2b).

In addition, it is worth noting that the spectral method only outputs an estimate of confusion matrices. We need post-processing to obtain an estimate of the true labels. Here we perform an E-step, which corresponds to calculating the posterior distribution of labels. As shown in Theorem 2 and Theorem 3, this procedure results in an optimal estimate of the true labels. An empirical evaluation of this procedure is plotted in Figure 1 as “Opt-D&S: 1st iteration”.

Assigned_Reviewer_17

We may have a somewhat different perspective on the spectral method and EM than is taken by the reviewer. Inferentially, we note that the spectral method is the method of moments (MoM) and EM is the method of maximum likelihood (ML). Both of these general approaches to inference require models for their analysis, in particular for analysis of consistency and rates of convergence, the focus of our paper, and the focus of much literature in theoretical statistics. Another kind of analysis is performance under model misspecification, which we agree is of significant interest, but which is beyond our current scope. Suffice it to say that the MoM and ML have advantages and disadvantages under various kinds of model misspecification and it will require a major effort of its own to explore these issues.

We want to clarify that both the spectral method and EM are necessary in the theoretical perspective we have presented. Without the spectral initialization, the EM output is heuristic. Without the EM algorithm, the spectral method cannot attain the optimal convergence rate. When they are combined, the success of the EM algorithm only requires a roughly accurate initialization, which overcomes the limitations of both. Please refer to the feedback to Reviewer 15 for more detail on this issue.

Empirically, we are happy to report the runtime of our algorithm in our revision.

As for the lower bound, both $\hat y$ and $\hat \mu$ are estimators that map the dataset to the target values. The infimum chooses the best possible estimator. Theorem 3 shows that both the required number of workers m and the error of estimating the confusion matrix are nearly unimprovable, for **any** value of n. The second part of constraint (9) (involving n) might be improvable. In real dataset we usually have n >> m. So we think that the optimality for m is more important than for n.

Assigned_Reviewer_37

We will provide more intuitive explanation to the impact of problem-specific constraints, and discuss when the conditions are satisfied by real problems.

The assumption that all confusion entries must be non-zero can be addressed by a simple argument. Suppose that the underlying model doesn't satisfy the assumption. Given any observed label, we replace it by a random label in {1,...,k} with probability $\rho$. In this modified generative model, every entry of the confusion matrix is lower bounded by $\rho/k$. If we run the proposed algorithm on this perturbed dataset, then the theory goes through and Theorem 2 holds. The random perturbation will make the constant $\bar D$ smaller than its original value, but the change is minor for small $\rho$.